# Fungal Secondary Metabolites and Small RNAs Enhance Pathogenicity during Plant-Fungal Pathogen Interactions

**DOI:** 10.3390/jof9010004

**Published:** 2022-12-20

**Authors:** Johannes Mapuranga, Jiaying Chang, Lirong Zhang, Na Zhang, Wenxiang Yang

**Affiliations:** College of Plant Protection, Technological Innovation Center for Biological Control of Plant Diseases and Insect Pests of Hebei Province, Hebei Agricultural University, Baoding 071001, China

**Keywords:** non-proteinaceous effectors, virulence, plant-pathogen interaction, cross-kingdom RNAi, extracellular vesicles

## Abstract

Fungal plant pathogens use proteinaceous effectors as well as newly identified secondary metabolites (SMs) and small non-coding RNA (sRNA) effectors to manipulate the host plant’s defense system via diverse plant cell compartments, distinct organelles, and many host genes. However, most molecular studies of plant–fungal interactions have focused on secreted effector proteins without exploring the possibly equivalent functions performed by fungal (SMs) and sRNAs, which are collectively known as “non-proteinaceous effectors”. Fungal SMs have been shown to be generated throughout the plant colonization process, particularly in the early biotrophic stages of infection. The fungal repertoire of non-proteinaceous effectors has been broadened by the discovery of fungal sRNAs that specifically target plant genes involved in resistance and defense responses. Many RNAs, particularly sRNAs involved in gene silencing, have been shown to transmit bidirectionally between fungal pathogens and their hosts. However, there are no clear functional approaches to study the role of these SM and sRNA effectors. Undoubtedly, fungal SM and sRNA effectors are now a treasured land to seek. Therefore, understanding the role of fungal SM and sRNA effectors may provide insights into the infection process and identification of the interacting host genes that are targeted by these effectors. This review discusses the role of fungal SMs and sRNAs during plant-fungal interactions. It will also focus on the translocation of sRNA effectors across kingdoms, the application of cross-kingdom RNA interference in managing plant diseases and the tools that can be used to predict and study these non-proteinaceous effectors.

## 1. Introduction

Plants have developed a broad spectrum of responses to counter pathogen invasion. Likewise, plant pathogens orchestrate a highly calibrated array of pathogenicity strategies in their quest to cause diseases [1]. The recent increased availability of fungal and plant genomes in the public domain has facilitated considerable progress in molecular plant–fungal interaction studies. Using genetic techniques, pathogenicity or virulence factors have been established, and the study of these factors has increased our understanding of the interactions between pathogens and their hosts. During interaction with their hosts, fungal plant pathogens secrete many proteins known as effectors which manipulate the physiology of the host or suppress the host’s immunity to promote infection [2,3]. Most studies on effectors have focused almost exclusively on secreted proteins, without exploring the possibly equivalent functions performed by fungal secondary metabolites (SMs) (chemical effectors) and sRNAs (sRNA effectors) which are collectively referred to as non-proteinaceous effectors [2,4,5]. Accumulating evidence has indicated that, pathogens use sRNAs (such as siRNAs and microRNAs) and SMs to manipulate host cell functions [6,7,8,9,10]. Fungal SMs and sRNAs have been show to manipulate host defense-related genes in the same was as proteinaceous effectors [5,8,11,12]. In general, SM and sRNA effectors are increasingly becoming important targets for studying the pathogenesis mechanisms of fungal pathogens [5,12]. Furthermore, these recently discovered SM and sRNA effector entities have been shown in a number of studies to be essential in manipulating host immunity and defense-related genes [2]. It is thus important to adopt new experimental methods to elucidate the in-planta biology of SM and sRNA effectors. This review reveals how fungal SMs and sRNAs enhance pathogen virulence during plant-fungal pathogen interactions. It will also discuss the translocation of these non-proteinaceous effectors across kingdoms, the application of RNA interference (RNAi) technology in managing fungal diseases of cereal crops and the tools that can be used to predict and study the SM and sRNA effectors.

## 2. Fungal SMs

Fungal SMs are not required for the growth and development of the fungus, but they have the potential to improve the pathogen’s fitness under certain conditions. Fungal SMs are often divided into polyketides, terpenes, non-ribosomal peptides and alkaloids on the basis of the primary enzymes and precursors that are involved in their biosynthesis [13,14,15]. They play a role prior to disease by shaping the plant microbial community, allowing producers to be fully adapted. The existence of fungal SMs, which have no discernible effect on the viability of the producer, raises issues about their potential influence on the environment [16]. SMs production by fungal pathogens and the presence of a host protein that is specifically susceptible to the corresponding toxin determines the ability of the pathogen to infect the host plant. Because host-specific toxin targets are encoded by plant genes, such genes can be referred to as dominant susceptibility genes [17]. It is generally known that mutualistic or pathogenic interactions between plants and fungal pathogens entail the simultaneous generation of molecular signals [11,18]. 

### 2.1. Fungal SM Biosynthetic Gene Clusters (BGCs) 

Toxins generated by fungi are likely merely a tip of the iceberg when it comes to non-proteinaceous effectors, and the precise roles of the majority of fungal SMs are yet unknown. Recent comparative genomics, molecular biology and bioinformatics studies revealed that genes that encode enzymatic activities to produce various fungal SMs are clustered and often found in close proximity to telomeres [13,19]. Genes found within a BGC are frequently co-regulated according to the function of the SM encoded by those genes [15], and the clustering of SMs is also important for epigenetic regulation of secondary metabolites expression. BGCs typically encode enzymes that are responsible for the biosynthesis of the metabolite backbone. These enzymes include non-ribosomal peptide synthetases, polyketide synthetases, fusions of polyketide synthetases and other enzymes that are involved in further modifications of the metabolite backbone [20]. In addition to this, certain BGCs contain genes that are involved in the transport of metabolites and/or genes that impart resistance to the action of the metabolite. Saprophytic fungi contain a significant repertoire of SM gene clusters in their genomes, typically with similarity to SM effector gene clusters of closely related pathogens [21,22]. However, biotrophic fungal pathogens must restrict the formation of SMs that are poisonous to their hosts [23]. In line with this, obligate biotrophic fungal pathogens *Blumeria graminis* f. sp. *hordei* (*Bgh*), *Melampsora larici-populina* (*M. larici-populina*), *Puccinia graminis* f. sp. *tritici* (*Pgt*), and *Puccinia triticina* (*Pt*), all exhibit a significantly reduced SM gene complement [24,25,26]. It was found that the last three species lack polyketide synthase genes and only have one non-ribosomal peptide synthetase gene, while *Bgh* has one polyketide synthase and one non-ribosomal peptide synthetase gene [23]. These findings suggest that there may be additional mechanisms linked with a biotrophic lifestyle besides the lower capacity of the production of SMs, such as the down-regulation of several SM biosynthetic pathways [23,27,28]. 

Findings from some transcriptomic studies reported that many fungal SM BGCs are expressed during particular plant colonization stages [29,30,31,32,33,34]. For example, the BGC encoding trichothecene, a *Fusarium graminearum* virulence factor, was upregulated during the infection of plants [15,35]. Moreover, during the infection of barley, wheat or maize (*Zea maize*), 41 BGCs of *F. graminearum* were expressed [31], and 11 *Zymoseptoria tritici* BGCs were expressed during the biotrophic infection stage of wheat leaves or at its early necrotrophic stage [5,30,36]. In *Colletotrichum higginsianum,* 14 BGCs were expressed during the penetration or biotrophic colonization of *Arabidopsis thaliana* leaves [34]. Some SM clusters that have been found in plant pathogenic fungi genomes show that more SMs may function as effectors throughout the infection process [37]. It was reported that some genes are specifically expressed during the infection process. For example, *Magnaporthe oryzae ACEA1* cluster is exclusively expressed in the appressoria during the penetration process [29,38], but they are not expressed in mycelium or spores later on in the infection phase. Moreover, another *M. oryzae* polyketide synthase-non-ribosomal peptide synthetase (PKS-NRPS) gene, *SYN8*, has the same expression pattern as *ACE1* and *SYN2* [17], demonstrating that this expression pattern is present in many SM clusters. It was concluded that during infection, half of the essential *M. oryzae* SM genes are either specifically expressed or upregulated, indicating an association between SMs and pathogenicity. These transcriptomic studies mainly focused on hemi-biotrophic fungal pathogens, and there are limited studies on the production and function of SMs in biotrophic fungal pathogens such as fungal rust pathogens and powdery mildews. Therefore, future research needs to explore expression of fungal SM BGCs during plant-biotrophic fungal pathogen interactions. 

### 2.2. Fungal SMs Enhance Pathogenicity during Plant-Fungal Pathogen Interactions 

Accumulating evidence indicates that fungal SMs serve as avirulence factors, host defense suppressors, and fungal cell wall hardening factors [5,11,39]. Fungal SMs are most effective during the early stage of infection (biotrophic phase), enhancing the fungus’ ability to penetrate and colonize its host without killing its host [2]. Fungal SMs can be host specific or non-host specific (Figure 1) and generate necrosis in plant tissue. However, some fungal SMs have functions linked to virulence that are not related to necrosis [1,27,40,41]. As long as the host plant has the relevant molecular target, such as a resistance gene product, SMs serving as host-specific effectors are thought to play an important role in pathogen virulence [1,42]. Paradoxically, SMs acting as non-host specific effectors have been widely regarded as critical components of pathogen arsenals, despite the fact that they may not be required for pathogenesis [23]. The majority of the fungal SMs have not been defined chemically, and the plants that they are intended to affect are still a mystery. The biological actions that have been reported to be caused by fungal SMs generated in-planta suggest that they have a broad range of plant cellular targets. Some fungi use high affinity iron chelator siderophores synthesized by NRPSs to scavenge environmental iron or to sequester cellular reactive iron [43,44]. These siderophores are essential for fungal growth and development, thus enhancing pathogenicity of various fungal pathogens. Cytochalasans, a diverse group of fungal PKS-NRPS hybrid metabolites, inhibit actin polymerization [45]. The production and transport of proteins are targets of a wide range of fungal SMs [5,46,47]. For example, the mycotoxin deoxynivalenol (DON), a member of the type B trichothecenes, produced by *Fusarium* spp., inhibits protein biosynthesis by binding to the ribosome, resulting in cell signaling, differentiation, reproduction and even teratogenicity disorders in eukaryotes [48,49,50].

A comparative transcriptome analysis of symptomatic and symptomless wheat tissues revealed a substantial induction of *TRI* genes in symptomless tissues, indicating that DON plays an important role in modulating host defenses and infection establishment [33]. Metabolite profiling of *F. graminearum* wild-type and the *tri5* deletion mutant in infected rachis nodes supports the function of DON in suppressing host defense-related metabolites [55]. DON was demonstrated to modulate programmed cell death (PCD) of host plant cells in a concentration-dependent way [56,57]. A higher concentration of DON may be produced by *F. graminearum* during infection to trigger hydrogen peroxide (H_2_O_2_) production by increasing the size of the hyphal colony. This results in further induction of PCD in wheat [56,57], and thus enhances its switch from biotrophy to necrotrophy [55]. Therefore, it can be unequivocally concluded that, during infection, the mycotoxin DON is produced as a sophisticated strategy of the fungal pathogen to circumvent and hijack the host plant’s defense system. 

*Cochliobolus* species were reported to produce host-specific toxins that enhance pathogen virulence. Victorin, a non-ribosomal peptide produced by *Cochliobolus victoriae*, is a virulence factor that enhances pathogenicity by inducing PCD during infection of only oat cultivars harboring susceptible genes [58,59,60]. It was also reported that victorin targets the plasma membrane and triggers PCD signaling pathways. HC-toxin, a non-ribosomal peptide produced by *Cochliobolus carbonum* induces histone hyperacetylation through the inhibition of histone deacetylases, during the infection of only maize varieties harboring susceptible genes [61,62]. Transcriptional activation of host plant defense genes is altered by such histone modifications, thereby enhancing pathogen virulence [62,63]. *Alternaria alternata* have various pathotypes that produce different host specific toxins that are active only on their corresponding susceptible hosts [64]. Some host specific toxins including destruxin which is produced by *Alternaria brassicae* are also essential for pathogens in susceptible host plants. *A. alternata* also produces AAL-toxin, an SM which enhances pathogenicity in tomato varieties harboring susceptible genes by inhibiting ceramide synthase. This will lead to free phytosphingosine and sphinganine accumulation followed by the disruption of plasma membrane [65]. Depudecin is another SM produced by *Alternaria brassicicola* which also enhances pathogenicity by inhibiting histone deacetylases; its role in pathogenicity is weaker than that of HC-toxin [66]. Tenuazonic acid produced by members of the genus *Alternaria* and other phytopathogenic fungi inhibits protein biosynthesis on ribosomes [67,68]. 

It was hypothesized that the *Colletotrichum graminicola* disease cycle is supported by monorden and monocillins in various ways, initially promoting biotrophic asymptomatic infection by inhibiting Hsp90 chaperons of R-proteins, and disrupting a maize hypersensitive response by enabling a switch to necrotrophy through suppression of basal plant defenses [69]. Fungal SMs such as ophiobolin and herbarumin enhance virulence by inhibiting calmodulin signaling which will disrupt plant regulatory networks [70,71]. It was also demonstrated that *Colletotrichum higginsianin* SM higginsianin B inhibits jasmonate-mediated plant defenses [72]. Two non-ribosomal octapeptides Fusaoctaxin A and B, which are biosynthesized by the gene cluster *fg3_54*, were found to be *F. graminearum* virulence factors [73,74]. Fusaoctaxin A alters the subcellular localization of chloroplasts in coleoptile cells and inhibits callose deposition in plasmodesmata during pathogen infection, thereby facilitating *F. graminearum* cell-to-cell penetration in wheat cells [73,74]. Table 1 summarizes the list and functions of some characterized fungal SMs that enhance pathogen virulence during plant-fungal pathogen interactions. Altogether, it can be concluded that fungal SMs are virulence factors that are integral players in the phytopathology canon. 

## 3. sRNAs—The Secret agents in Plant-Fungal Pathogen Interactions 

Plant immune responses are tightly regulated by an array of immunity-associated regulators such as sRNAs and some transcription factors [82]. Based on their biogenesis and structural features, sRNAs can be classified into three categories: short-interfering RNAs (siRNAs), dicer-independent microRNAs (miRNAs) and dicer-independent piwi interacting RNAs (piRNAs) [10,83,84,85]. The fundamental sRNA pathway components and other various sRNAs function as critical gene expression regulators to fine-tune the immunity of some cereal plants such as wheat and rice against pathogen invasion [82]. Normally, when a pathogen attacks its host, these sRNAs are either upregulated or downregulated in order to inhibit expression or to release suppression of their targets [6,86]. Thus, plant endogenous sRNAs and sRNA pathway components play key roles in regulating and fine-tuning host immune responses to pathogens such as fungi, bacteria, and oomycetes [87]. Accumulating evidence indicates that sRNAs produced by fungal pathogens can function as effector molecules, modulating host gene expression as a counter-defense mechanism (Table 2) [6,8,10,88,89,90,91]. 

### 3.1. Cross-Kingdom RNAi during Plant-Fungal Pathogen Interactions 

During pathogen infection, the host’s sRNA performs an endogenous role by regulating gene expression in order to maintain a healthy balance between plant development and immunity [106,107,108]. sRNAs can direct the transcriptional and post-transcriptional silencing of gene expression, and this phenomenon is known as RNA interference (RNAi). Post-transcriptional gene silencing is a mechanism through which plant miRNAs contribute to resistance by regulating the expression of defense-related genes [109,110]. The phenomena of cross-kingdom RNAi occurs when gene silencing is induced between unrelated species from different kingdoms, like a plant host and its interacting pathogen (Figure 2). It necessitates the translocation of a gene-silencing trigger from a donor into an interacting recipient. Several studies have reported interactions with other species in plant and animal systems through cross-kingdom RNAi [87,111,112]. sRNAs generated by pathogens and parasites, on the other hand, may also translocate into host cells and induce host gene silencing [88,111,112,113,114,115]. This implies that sRNA transfer is bidirectional; plant-derived sRNAs serve as defense weapons to disrupt fungal pathogenicity genes, while pathogen-derived sRNAs act as offensive weapons to suppress host plant defense mechanisms (Table 2) [5]. 

Since the discovery of RNAi in *Neurospora*, sRNAs from numerous fungal species have been studied [116]. The most well-known example of cross-kingdom RNAi from a plant to its interacting pathogen is HIGS, which occurs when a plant-produced RNAi signal triggers the silencing of a pathogen gene [117,118]. RNase III-like endonucleases known as Dicers produce sRNAs from hairpin-structured or double-stranded RNA [119]. The mature sRNAs are loaded into AGO proteins to form the RNA-induced silencing complex (RISC) [6,120]. The RISC is responsible for silencing genes that contain sequences complementary to sRNAs. By using the component of the host RNAi machinery known as AGO1, the transfer of *B. cinerea* sRNA into *Arabidopsis* cells silenced the host’s immune genes [88]. Fungal sRNAs can suppress host plant immunity by interfering with the RNAi pathways of the host [8,88,89]. During tomato and *Arabidopsis* infection, the most prevalent sRNAs that function as effectors factors to enhance pathogen virulence are *Bc*-siR3.1, *Bc*-siR3.2, and *Bc*-siR3.5 which target the host *mitogen-activated protein kinases MPK1*, *MPK2* and *MPKKK4*, *peroxiredoxin* (*PRXIIF*), and *cell wall-associated kinase* (*WAK*), respectively [88]. These pathogen-derived sRNAs target components of host plant immunity such as oxidative burst and signal transduction pathways; hence, silencing of these targets will enhance pathogen virulence and compromise resistance to the fungal pathogens [88]. In support of the hypothesis that sRNAs enhance *B. cinerea* virulence, a *dcl1 dcl2 B. cinerea* double mutant that could not produce *Bc*-sRNAs exhibited attenuated virulence, whilst a *dcl1* or *dcl2* single mutant still produced sRNAs to sustain pathogenicity on host plants [88,121]. *B. cinerea* also delivers Bc-siR37 into the host cells, which targets host *PMR6*, *WRKY7* and *FEI2*, leading to suppression of host immunity [89]. 

A group of fungal in-planta secreted sRNAs was also identified from the sequencing of sRNAs from *Sclerotinia sclerotiorum* during infection of *Arabidopsis* and *Phaseolus vulgaris* [122]. The pathogen-derived sRNAs were predicted to target quantitative disease resistance-associated genes of the host and suppress host plant immunity [122]. Mutations of two sRNA targets that encode kinase genes *SERK2* and *SNAK2* enhanced pathogen virulence and compromised host plant resistance, indicating that the sRNAs’ targets are involved in disease resistance [122]. Analysis of *Pst*-infected leaves established that *Pst* is capable of suppressing the host’s defense and immunity genes as well as its endogenous genes by producing many sRNAs [7]. When a fungal infection occurs, the miRNAs involved in disease response either up- or down-regulate the expression of their target genes [123,124]. For example, there was a substantial increase in *miR1138* levels in bread wheat infected with *Pgt* (62G29-1) [124]. *miR393*, *miR444*, *miR827*, and *miR2005* were upregulated in wheat (*T. aestivum* L.) following *B. graminis* infection [125]. The open reading frame or untranslated regions of certain genes are targeted by miRNAs, which have the ability to inhibit the translation of those genes [126].

Fungal sRNAs can target and silence plant transcripts involved in defense, but sRNAs from plants can target and silence transcripts produced by pathogens [127,128]. A novel *Pst* miRNA (*Pst*-milR1) participates in cross-kingdom RNAi events in wheat by binding the *pathogenesis-related 2* (*PR2*) gene, which may suppress the host-mediated defense mechanism in its counter defense. Silencing of the *Pst*-milR1 precursor using host induced gene silencing resulted in reduced *Pst* virulence and increased wheat resistance to the *Pst* isolate CRY31. Therefore, *Pst*-milR1 is a key pathogenicity factor in *Pst*, which functions as an effector to suppress host immunity [8]. Computational prediction of targets using a common set of sRNAs and *Pt* mil-RNAs (pt-mil-RNAs) within *Pt* and wheat found that the majority of the targets of *Pt*-derived sRNAs were repetitive elements in *Pt,* whilst in wheat the target genes were revealed to be involved in various biological processes including defense-related pathways [10]. The sRNAs’ targeted genes are involved in disease resistance, metabolic processes, transporter, and apoptotic inhibitor activities. This was the first study to report the discovery of new sRNAs found in *Pt* [10]. Expression validation studies performed on twenty individual *Pt*-sRNAs and two pt-mil-RNAs, namely pt-mil-RNA1 and pt-mil-RNA2, showed evidence of their possible role in pathogenicity or virulence on the host. pt-mil-RNA1 and pt-mil-RNA2 were both found to suppress wheat defense response to *Pt* by targeting wheat transcription factor TCP14, cytochrome b5 reductase and elongation factor 2 [10]. *Fg-sRNA1* produced by *F. graminearum* targets and silences wheat *TaCEBiP* (*Chitin Elicitor Binding Protein*), a pattern recognition receptor gene (Figure 3) [90]. *F. oxysporum* f. sp. *lycopersici* produces *Fol-milR1*, an sRNA effector that suppresses host immunity by targeting the tomato protein kinase *SlyFRG4* via *AGO4a*, thus providing a novel pathogenicity strategy to achieve infection [91]. 

Centromeric sRNAs associated with genome-wide hypermethylation were induced by the stem rust pathogen *Pgt* during late infection stages [129]. Although *Pgt*-derived sRNAs maybe be used by the pathogen for silencing target host genes, endogenous functions of these sRNAs were also discovered during infection [129]. A recent study established that, upon infection, particular wheat 24-nt heterochromatic siRNAs (hc-siRNA) were repressed, whereas specific 25-nt rRNA and tRNA fragments were significantly upregulated. Transcripts encoding a ribosomal protein and a glycosyl hydrolase effector in the fungal pathogen were cleaved by wheat sRNAs [130]. Long inverted repeats in the vicinity of protein coding genes in fungi gave rise to the development of miRNA-like and phased 21-nt sRNAs. sRNAs produced by fungi targeted not only native transcripts, such as transposons and kinases, but also transcripts from other kingdoms, such as a wheat nucleotide-binding domain leucine-rich repeat receptor (NLR) and several families of transcription factors involved in defense. This research provides new insights into host-microbe coevolution and opens up promising ways to improve biotechnology to control pathogens [130]. 

Accumulating evidence shows that miRNAs serve crucial roles in regulating the expression of their target genes accurately and effectively during the interactions between rice and *M. oryzae*. Understanding the functions of rice miRNAs is crucial for managing rice blast. *miR398b* coordinates various pathways to increase the accumulation of H_2_O_2_ through numerous *Superoxide Dismutase* (*SOD*) family genes [131], thereby positively regulating rice defense responses to *M. oryzae* [110]. *miR166k-miR166h*, *miR160a*, and *miR7695*, positively regulate [110,132,133], whereas *miR444b.2*, *miR164a*, *miR319b*, *miR169*, *Osa-miR439*, and *miR396* negatively regulate rice resistance to *M. oryzae* [97,99,100,103,104,110]. *Osa-miR167d* is a member of a conserved miRNA family that functions in developmental and stress-induced responses by regulating the expression of genes encoding *auxin responsive factors* (*ARFs*). It was demonstrated that *Osa-miR167d* downregulates *ARF12*, a component of rice immunity, to enhance *M. oryzae* infection. Therefore, the *Osa-miR167d-ARF12* regulatory module may be helpful in enhancing resistance to blast diseases [95]. 

It was previously shown that plants may transfer miRNAs to the fungal pathogen *Verticillium dahliae*, to activate antifungal RNAi. It was recently shown that *V. dahliae* may secrete an effector to the plant nucleus, where it can interfere with the nuclear export of AGO1-miRNA complexes, thereby preventing antifungal RNAi and enhancing pathogen virulence. These findings revealed an antagonistic mechanism through which fungal pathogens can manipulate plant sRNA function to thwart antifungal RNAi immunity [134]. Zhu and colleagues proposed that *VdSSR1* is translocated from *V. dahliae* cells to the nucleus of the host plant, probably through a noncanonical secretion pathway. *VdSSR1* localized in the nucleus inhibits the nuclear export of AGO1-miRNA complex and mRNAs by sequestering ALY adaptors and inhibiting them from associating with the UAP56-TREX complex [134]. In the plant-*V. dahliae* system, *VdSSR1* acts as a general suppressor of plant miRNAs, including trans-kingdom *miR159* and *miR166*. Attenuated translocation of accumulated cytoplasmic *miR166* and *miR159* to *V. dahliae* cells ultimately suppresses trans-kingdom silencing of pathogen virulence genes and promotes fungal infection. *VdSSR1*-mediated suppression of the mRNA export of specific defense genes may potentially contribute to the increased virulence in plants [134]. Based on these studies, it can be concluded that sRNAs can perform a wide range of functions, including manipulating host machinery in the same way as classic proteinaceous effector molecules. Altogether, these studies demonstrate that pathogen-derived sRNAs can be translocated into the host and function as effectors to suppress host defense genes.

### 3.2. Applications of Cross-Kingdom RNAi Technology 

Several studies have demonstrated RNAi-based fungal pathogen management with an average plant disease resistance of approximately 60% [135]. Cross-kingdom RNAi was initially studied to generate disease resistance in barley and wheat against *B. graminis*, the powdery mildew fungus, using HIGS, an RNAi-based approach [136]. The HIGS technique for controlling pathogens including fungi and viruses, among other plant pests, was developed with help from plant immune system’s RNA silencing machinery against viruses [136,137,138]. To silence pathogen target genes, the HIGS employs RNAi by producing sequence-specific dsRNAs in the host plant. A hairpin-structure dsRNA construct which targets a specific gene is transformed into the host plant. dsRNAs and siRNAs produced by the transgenic plant are taken up by corresponding plant pathogens during host-pathogens interactions. These siRNAs target and degrade pathogen mRNAs, hence protecting the host plant against pathogen infection [118,139]. HIGS technology has been widely adopted in plant breeding programmes as an efficient approach to enhance plant defense responses to pathogens. 

The HIGS approach has been broadly utilized to manage wheat and barley fungal pathogens like the *Puccinia* species, *Fusarium* species, and *B. graminis* [136,140,141,142,143,144,145,146]. Obligate biotrophic fungal pathogens, *Blumeria graminis* f. sp. *tritici* (*Bgt*) and *Bgh* cause severe powdery mildew in wheat and barley, respectively. *Bgh* effector *Avra10* and *MLa10*, a disease resistance gene in barley, were used to perform a proof-of-concept for HIGS [136]. Avra10 is an important *Bgt* pathogenicity factor. However, recognition of Avra10 by barley MLa10 triggers hypersensitive response in the host plant, resulting in suppression of biotrophic pathogen invasion [136,147]. *Avra10* silencing on barley leaves exhibited attenuated pathogen development without *MLa10*, but not with it [136]. The HIGS approach was also used to screen fifty *Bgh* haustoria-associated effectors, and the silencing of eight of them significantly reduced pathogen virulence in barley [141]. HIGS technique was further used in transgenic wheat plants to combat the rust pathogens *Pst*, and *Pt* [144,145,146]. Zhu and colleagues generated transgenic wheat-derived dsRNAs targeting *PsFUZ7*, a MAP kinase, which contributes to *Pst* pathogenicity by regulating the morphology and development of hyphae [144]. Strong resistance to *Pst* infection was exhibited by transgenic lines stably expressing dsRNA constructs by degrading *PsFUZ7* transcripts, leading to the suppression of pathogen growth and development [144]. 

It was also shown that, in transgenic wheat lines expressing target dsRNAs which disrupt the *Pst* cAMP-PKA chain pathway, silencing of a *Pst* protein kinase A (PKA) catalytic subunit *(PsCK1*) inhibits wheat stripe rust. Throughout the T3 and T4 generations, these transgenic lines maintained a high level of resistance to wheat stripe rust [146]. Based on these results, *PsCPK1* and its homologous genes are promising targets for developing transgenic plants with long-lasting resistance to stripe rust fugus *Pst*. HIGS induced by barley stripe mosaic virus (BSMV)-VIGS (virus-induced gene silencing) was established to be a robust, high-throughput approach for the functional analysis and validation of rust fungi candidate genes involved in pathogenicity through quantitative estimation of infection-related traits [148]. It was discovered that wheat and barley plants that generated dsRNA or anti-sense RNA fragments, both of which were intended to alter gene expression in the fungus, silenced the fungal genes [136]. Hairpin RNAi constructs with sequence similarity to *MAP kinase1* (*PtMAPK1*) or *Cyclophilin1* (*PtCYC1*) silenced the respective fungal genes and conferred resistance to leaf rust pathogen *Pt* [145]. It was revealed that, the silencing signals in the *Puccinia*-wheat pathosystems are most likely host cell-derived siRNA molecules [149]. Using site specific analysis, a recent study revealed putative cross-kingdom sRNAs, tRNA and rRNA fragments, and some signs of fungal phasing in the barley-*Bgh* interactions [150]. This was the first research to report on phased short RNAs (phasiRNAs) in *Bgh*, a trait normally associated with plants that may be involved in the post-transcriptional regulation of fungal coding genes, pseudogenes, and transposable elements [150].

Fusarium head blight (FHB) caused by the fungi *F. graminearum* and *F. culmorum* is another destructive disease of wheat and barley [151,152]. Koch and colleagues generated transgenic *Arabidopsis* and barley plants expressing dsRNA against three *CYP51* paralogous genes that are essential in the ergosterol biosynthesis pathways [140]. Inhibitors of sterol demethylation, which act on the *CYP51* paralogs, are the most extensively used systemic fungicides for controlling fungal pathogens. Increased resistance to *F. graminearum* infection was shown after the transgenic expression of the three paralogous genes in two different plant systems [140]. The expression of three hairpin RNAi constructs in wheat transgenic lines also silenced *chitin synthase* (*Chs*) *3b*, a *F. graminearum* pathogenicity factor, and these lines exhibited firm and consistent resistance to Fusarium seedling blight (FSB) and FHB over T3-T5 generations [142]. Fungal infection on wheat ears and seedlings was significantly reduced by the expression of dsRNAs with sequence homology to *F. graminearum Chitin synthase Ch3b* [142]. Transient HIGS in wheat, and targeting an important housekeeping gene in *Bg* led to considerable decrease in virulence during early infection stages [108].

Rice blast is one of the most devastating rice diseases caused by the pathogenic fungus *M. oryzae* [153]. Zhu and colleagues studied the effects of silencing three virulence-related genes, ABC transporter *MoABC1*, membrane-bound adenylate cyclase *MoMAC1*, and mitogen-activated protein kinase *MoPMK1*, during the interactions between rice and *M. oryzae* [154]. The resistance of rice to blast was enhanced when three BSMV silencing vectors targeting *MoPMK1*, *MoMAC1*, and *MoABC1* were inoculated into the host plant at once. Furthermore, the development of disease was inhibited by the silencing of the individual pathogen genes [154]. Recently, a *M. oryzae* strain was subjected to an in vitro sensitivity assay with artificial siRNAs (asiRNAs) targeting four candidate genes involved in fungal virulence: *MoSSADH* encoding succinic semialdehyde dehydrogenase, *MoAP1* transcription factor, and its downregulated genes *MoAAT* encoding aminobutyrate aminotransferase, and transcription factor *MoSOM1* [155]. Feeding the fungal strain siRNAs specific to *MoAP1* inhibited growth and pathogenicity, whereas siRNAs specific to *MoSSADH*, *MoSOM1*, and *MoAAT* did not. Additional in vivo HIGS testing showed that transgenic rice lines expressing an RNA hairpin targeting *MoAP1* had enhanced resistance to eleven *M. oryzae* strains [155]. Altogether, cross-kingdom-RNAi has the potential to supplement existing pathogen management strategies, for instance, by widening the resistance spectra of host resistance genes. Furthermore, using cross-kingdom-RNAi to exploit naturally occurring RNA exchanges may open up new avenues for crop improvement through genetic engineering and classical breeding.

## 4. Translocation of Non-Proteinaceous Effectors across Kingdoms: Extracellular Vesicles as Mediators of Infection

In effector biology, the mechanisms through which fungal effectors, particularly sRNA and SM effectors, are transported into cells of the host plant to their targets remain a matter of speculation. Accumulating evidence from preliminary studies suggests that genetic material may be transferred from the host plant to the infecting fungal pathogen cell through exosomal biogenesis pathways [149,156,157,158]. At fungal penetration sites, multivesicular compartments aggregate around fungal haustorial complexes in the host cytoplasm, allowing differentiated vesicle trafficking across the plant-pathogen cellular interface to occur anterogradely, and possibly retrogradely. These multivesicular bodies consist of several intraluminal vesicles, which are discharged extracellularly as exosomes into the paramural region after fusion with the plasma membrane [149]. Multivesicular body-like compartments were reported to be involved in trafficking processes at intercellular channels known as gap junctions, nanotubes, and even the internalization of plasma membrane sections by neighboring cells [159]. siRNA species generated in the host silencing donor were suggested to be transmitted to the fungal recipient through an exocytic/endocytic exchange process at the haustorial interface [149]. Exosomes and plasma membrane-budded microvesicles have both been identified as extracellular vesicles (EVs) that are secreted by plant cells and found in the cells of fungal pathogens [5,160]. 

EVs serve as mediators of infection and defenses during plant–fungal pathogen interactions (Figure 2). Active extracellular vesicular transport, passive transport via trans-cell wall diffusion, binding and internalization through membrane-associated receptors, and other trans-membrane pores or channels are all possible sRNA trafficking mechanisms across the plant-fungal interface [5,111,117,161,162,163,164,165]. A diverse collection of plant sRNAs, including miRNAs and siRNAs, are selectively loaded into the EVs of plant cells [85,163]. It was suggested that fungal sRNAs delivery is facilitated by EVs, similar to the suggested plant-extracellular vesicle-mediated sRNA transport [166]. To test this hypothesis, EVs isolated from various fungal pathogens including *F. oxysporum* [167,168], *F. graminearum* [168,169], *Z. tritici* [170], and *Ustilago maydis* [171], were established and this laid a foundation for future study of cross-kingdom RNA transport in plant-fungal pathogen interactions [172]. The secreted EVs included a variety of membrane-trafficking proteins and numerous proteins for substrate transport, indicating that EVs might serve an important role in RNA trafficking [162]. Altogether, accumulating evidence points to the idea that fungal EVs are a viable method of transporting pathogen effector proteins, sRNAs and SMs into host plant cells, and their packaging in membrane-bound compartments protects them from degradation by the host enzymes and dilution by water in the plant apoplast [5].

## 5. Tools for the Prediction and Study of SM and sRNA Effectors 

In effector biology, genomes, transcriptomes, proteomes, and metabolomes are mined to facilitate the discovery of potential effector genes for molecular or cellular biology, biochemistry, and reverse genetics (Figure 4). Future studies need to focus on the development of integrated approaches for the molecular and functional characterization of fungal SM and sRNA effectors during interactions with their host plants. Deletion mutants are usually studied for pathogenicity or symbiosis [5]. The most significant obstacles in the generation of deletion mutants in fungi continue to be transformation and homologous recombination (Figure 4). This strategy has various difficulties when used for SM and sRNA effectors; therefore, new experimental approaches are needed to overcome them. Phylogenetics and comparative genomics analyses should be performed before experimental research because they are particularly informative since the number of fungal genomes and documented SM pathways is increasing. In silico studies may provide novel insights into the organization of conserved gene clusters, as well as their limits and evolutionary history. These kinds of approaches are extremely useful in locating gene clusters that play a role in the production of SMs that have been characterized in other fungal species. They also make it possible to predict the production of compounds that are either identical or related to those produced by specific fungal species. The increased availability of fungal genome sequences and next-generation genomic technologies enables the assessment of SM gene clusters in an individual fungus. RNA-Seq has revolutionized transcriptome profiling and is utilized to study SM gene cluster expression during infection. RNA-Seq can simultaneously quantify transcripts from many organisms, making it ideal for studying plant-pathogen interactions. Manipulating strain-unique SM genes involved in host-specific pathogenicity facilitates plant-fungal pathogen interactions research. Using BGC expression in heterologous hosts such as *Saccharomyces cerevisiae* or *Aspergillus* spp. may help to overcome functional redundancy and in-planta detection limitations. SMs from plant pathogenic fungi have primarily been evaluated using phytotoxicity tests. The utilization of chemical genetic screenings may also discover actions against phytohormone signaling pathways and PTI responses. For high-throughput chemical screening, this technique may utilize *A. thaliana* transgenic lines that express reporter genes in 96-well microplates [173]. 

Patterns of gene expression and regulation may be used to decipher the complicated bidirectional interaction between pathogen and host cells [121]. Using RNA-seq on both the pathogen and the host is an effective way to examine both sides of this relationship [174,175]. The recently discovered CRISPR Cas13 system can be used to study sRNA effectors, for instance, through the inactivation or localization of fungal sRNAs [176]. Recent studies have shown that extracellular vesicles play significant roles in host defense and pathogen virulence as well as being essential tools for communication between plants and pathogens. To induce the silencing of fungal genes essential for pathogenicity, plant cells secrete extracellular vesicles containing sRNAs into fungal cells. Transmission electron microscopy following ultra-rapid cryofixation showed EVs in *Golovinomyces orontii* extrahaustorial matrix [177]. EVs produced by apoplastic pathogens may be detected from plant washing solutions [178]. Such fluids likely include plant and fungal EVs, making it difficult to determine their source of origin. To find out if plant pathogen EVs carry RNAs that are functional inside host plant cells, more research must be done, including the biogenesis of EVs and how specific molecules are sorted and directed towards them. 

## 6. Conclusions and Future Perspectives 

Many plant–fungal interactions now have genome and in-planta transcriptome data, which has aided in characterizing the repertoire of fungal proteinaceous and non-proteinaceous effectors expressed during plant infection, when biomass accounts for a very small portion of the infected host plants. Therefore, the identification of SM and sRNA effectors will be very challenging and needs high sensitivity techniques to detect them. Such studies involve molecular, genetic, and more complex bioassays than pathogenicity tests. Most uncharacterized SM and sRNA effectors are produced during the penetration and initial infection stages. The diversity of fungal SM genes among species suggests that gene products, especially those related with distinct genomic areas, may influence pathogenic lifestyle. Metabolomics approaches are also being used to study the biochemical complexity of fungal-host plant interactions. It is interesting that only around 25% of the fungal SM gene clusters have already been functionally characterized, despite the fact that SMs play essential roles in the virulence and lifestyle of fungal plant pathogens. Fungal SMs have intriguing biological activities, but their contribution to pathogenicity is often overlooked, possibly because of functional redundancy. An integration of genetic and biochemical approaches is required for the construction of single and multiple mutants and the identification of the structures of corresponding SMs. 

Cellular stress responses and plant immunity are tightly regulated by both plant host endogenous sRNAs and pathogen-derived sRNAs. Many distinct types of immune-regulatory sRNAs have been discovered, and they are differentially regulated in response to pathogen infection. Some of them may translocate into species with which they interact, resulting in cross-kingdom RNAi. Based on microbiome studies, comparative genomics and sRNA deep sequencing will promote the investigation of critical cross-kingdom sRNAs implicated in plant-fungi interactions, unraveling novel efficient targets for crop protection techniques such as HIGS. The integration of HIGS with the utilization of a novel multi-transgene stalking toolkit comprised of marker-excision cassettes will optimize the silencing of many pathogens using several RNA constructs that will specifically target distinct genes in pathogens. Furthermore, transient reprogramming based on a variety of viral vectors has been utilized to initiate changes in agronomic traits including plant height, drought tolerance or flowering time [179]. In a similar vein, RNAi based on viral vectors can also be developed to effectively manage pandemics caused by plant fungal pathogens. Future plant protection will rely heavily on RNAi-based technologies. 

Although cross-kingdom RNAi has been shown in several instances such as HIGS, the mobile RNAi triggers and the mechanisms and pathway(s) of RNA transport remain elusive. Data that can help explain the basic mechanisms of sRNA translocation across plant and fungal cells are still required. It is also unknown how sRNAs are sorted and transported to target cells. Furthermore, the complex biogenesis pathways and uncertain roles of fungal sRNAs suggest intriguing issues for RNAi fundamental research. For example, how can a fungal RNAi system distinguish between endogenous and external sRNAs, which then serve roles in several signaling pathways? Answers to these and other cross-kingdom communication concerns will not only improve our knowledge of plant disease processes, but also assist in the establishment of effective novel disease-control tools. We anticipate that well-designed genetic, biochemical, and cell biology assays will provide insights into the cross-kingdom RNAi signal conveyance process. Furthermore, understanding the evolution of cross-kingdom RNA trafficking and how it affects host-pathogen interactions will need further research into the specific molecular processes regulating this process. As miRNAs have emerged as key regulators of host and pathogen gene expression, protecting agricultural plants from pathogens has increasingly become an issue of concern. In order to develop innovative ways for preventing pathogen infection in agricultural plants and increasing crop yield, further research into the miRNA-mediated mechanism in plant-pathogen interactions is required. miRNAs have the potential to serve as excellent biomarkers for identifying traits associated with disease resistance in breeding programmes. More research on miRNA cross-kingdom transfer is needed to fully comprehend the roles played by miRNAs in host-cell gene silencing and host pathogen trans-regulation of genes to develop innovative ways for managing pathogen infection in cereal crops to increase crop yield. Further research on cross-kingdom miRNA transfer would aid in gaining a better comprehension of miRNAs in gene silencing in the host plant and trans-regulation of genes in pathogens. Understanding the molecular mechanisms of mobile sRNA selection at various stages of development and in relation to environmental variables is a potential next step in sRNA research. 

## Figures and Tables

**Figure 1 jof-09-00004-f001:**
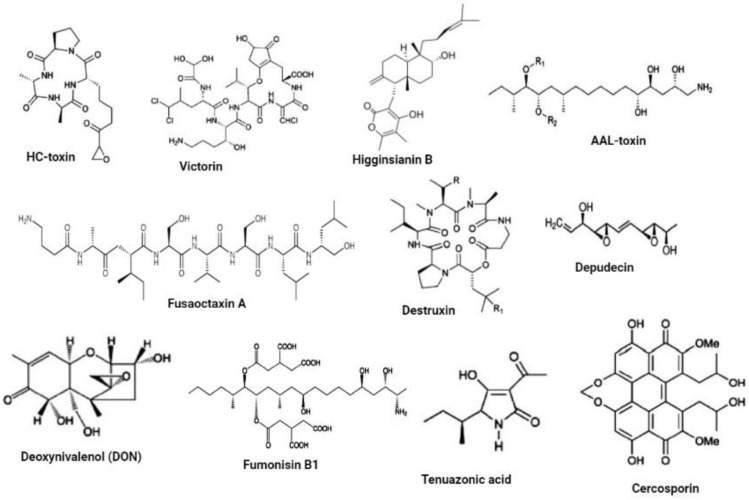
Fungal SM effectors. Host-specific toxins include HC-toxin, victorin, higginsianin B, AAL-toxin, Fusaoctaxin A, destruxin, depudecin. Host non-specific toxins include DON, Fumonisin B1, tenuazonic acid, and cercosporin. Adapted from [51,52,53,54].

**Figure 2 jof-09-00004-f002:**
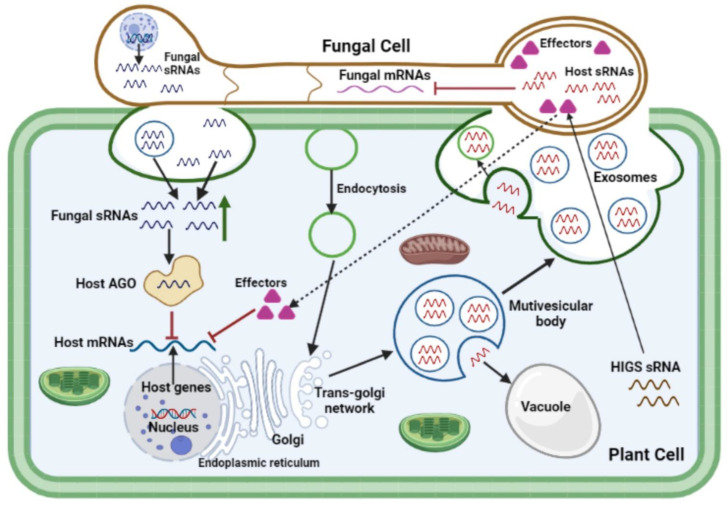
Cross-kingdom RNAi and vesicle trafficking during plant-fungal pathogen interactions. Fungal and plant sRNAs trigger cross-kingdom RNAi during plant-pathogen interactions. Fungal sRNAs translocate into plant cells and hijack the host plant Argonaute (AGO) protein of the RNAi machinery to suppress host plant immune response. The fungal sRNAs are upregulated upon infection (indicated by green arrow). Host cells also can deliver sRNAs into pathogen cell, either host induced gene silencing (HIGS) sRNAs or endogenous sRNAs, to target virulence genes and other essential pathogen genes. The generation of multivesicular bodies and release of exosomes at the site of pathogen invasion is part of the host penetration resistance pathway. Among other molecules, the putative exosomes contain sRNAs that can target vesicle trafficking components of the pathogen. Exosomes can also inhibit fungal growth and stall further ingress. The production of pathogen-derived sRNAs that may target and silence host genes can be inhibited by this form of host plant immunity. The fungal pathogens also secrete proteinaceous effectors through the haustorium into the host cells to suppress the host immunity genes, thereby causing disease. How fungal pathogens transport proteinaceous effectors and sRNAs into their host cells is still elusive. On the other hand, plants secrete extracellular vesicles to transport host sRNAs into pathogens to silence fungal genes involved in pathogenicity. Passage of host sRNAs through the haustorial cell wall, either active or passive, occurs and once inside the fungal haustorium the silencing molecules trigger RNAi of their mRNA targets, and may act as primers in the fungal silencing pathway, leading to the generation of systemic silencing signals. Cell structures are not drawn to scale.

**Figure 3 jof-09-00004-f003:**
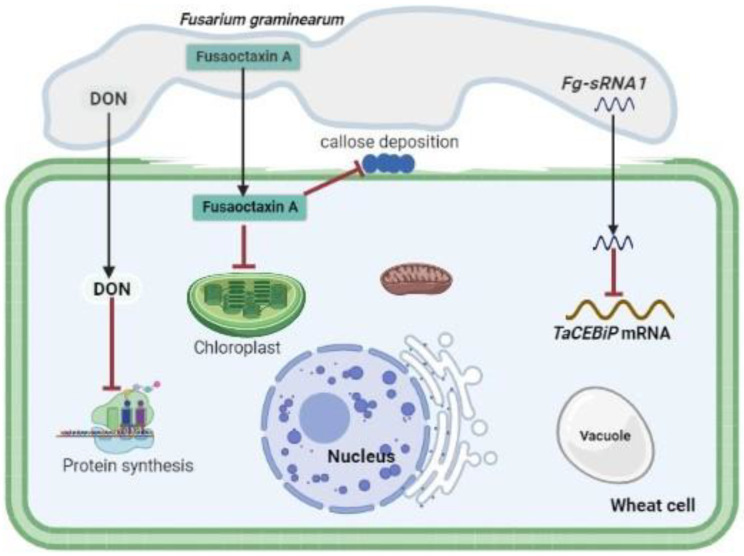
Mode of action of *F. graminearum*-secreted pathogenicity factors during *F*. *graminearum*-wheat interaction. The sRNA effector *Fg*-sRNA1 contributes to virulence by silencing wheat defense-related *TaCEBiP*. Fungal toxin DON inhibits protein biosynthesis by binding to the ribosome. The fungal toxin Fusaoctaxin A changes the subcellular localization of chloroplasts in the coleoptile cells and prevents callose accumulation in plasmodesmata during pathogen infection, facilitating the cell-to-cell invasion of *F. graminearum* in wheat tissues.

**Figure 4 jof-09-00004-f004:**
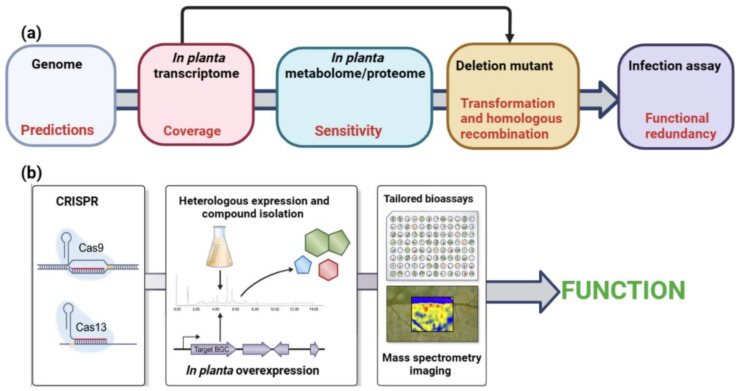
Integrated approaches to elucidate the biological functions of fungal SM and sRNA effectors during plant-fungal pathogen interactions. (**a**) The most common approach begins with genome mining integrated with in-planta transcriptome analysis (and/or in-planta proteome/metabolome analysis), which gives potential candidate genes to delete and test for a function in plant colonization. Following these functional tests, plant targets are identified. Each of these steps has bottlenecks (text in red), which necessitate the consideration of complementary or alternative options. Improved bioinformatic prediction of gene composition is critical for secondary metabolite gene clusters in particular. In-planta-omics approaches suffer from fungal material dilution in a complex plant sample. Transformation and homologous recombination continue to be the most significant barriers to the generation of deletion mutants in fungi. Infection assays are often sensitive enough to detect only significant contributions to the infection process. (**b**) Combinatorial genetic validation employing clustered regularly interspaced short palindromic repeats (CRISPR)-Cas9 and Cas13 technologies opens new opportunities for studying the biological functions of fungal SM and sRNA effectors. Heterologous production or enhanced in-planta synthesis of fungal secondary metabolites allows their chemical characterization and subsequently determination of their function using tailored bioassays and spatial distribution in infected plant tissue. Adapted from [5].

**Table 1 jof-09-00004-t001:** Fungal SM effectors involved in manipulation of host immunity.

Name of SM	Origin	Target and Function	Reference
HC-toxin	*C. carbonum*	Infects only corn varieties harboring the gene susceptible to HC-toxin. It inhibits histone deacetylases which alters the transcriptional activation of defense-related genes.	[61,62,63]
Victorin	*C. victoriae*	Infects only oat cultivars harboring a gene susceptible to victorin. It targets a resistance protein to trigger programmed cell death.	[58,59,75]
Higginsianin B	*C. higginsianum*	Inhibits jasmonate-mediated plant defenses.	[72]
AAL-toxin	*A. alternata*	Essential for the pathogen virulence on susceptible tomato cultivars. It causes membrane rupturing as free phytosphingosine and sphinganine accumulate due to inhibition of ceramide synthases.	[65,76,77]
Fusaoctaxin A	*F. graminearum*	Facilitates pathogen cell-to-cell penetration in wheat cells by altering the subcellular localization of chloroplasts in coleoptile cells and blocking the deposition of callose in plasmodesmata during pathogen infection.	[73]
Destruxin	*A. brassicae*	It is essential for pathogenicity and targets its corresponding susceptible gene in the host.	[64]
Depudecin	*A. brassicicola*	Modulates host plant immunity by acting on chromatin modifications and interfering with regulatory networks	[66]
Deoxynivalenol (DON)	*F. graminearum*	Binds to the ribosomes, thereby inhibiting protein biosynthesis. It triggers H_2_O_2_ production by enlarging the hyphal colony, which then induces PCD in wheat, and enhances its switch from biotrophy to necrotrophy.	[48,55,57]
Fumonisin B1	*Fusarium verticillioides*	Essential for the pathogen virulence. Inhibits ceramide synthase.	[77]
Tenuazonic acid	*M. oryzae*	Enhances pathogen virulence by forming complexes with metals which facilitates rapid spread of the pathogen on rice leaves.	[78,79]
Cercosporin	*Cercospora* ssp. and other phytopathogenic fungi	Serves as a virulence factor by absorbing light to generate oxygen radicals that will damage proteins, nucleic acids and lipids. This results in the leakage of these nutrients into the intracellular space which will make them available to the fungal hyphae, thus enhancing pathogen invasive growth and proliferation.	[80,81]

**Table 2 jof-09-00004-t002:** Fungal sRNA effectors and their target genes in cross-kingdom interactions.

sRNA	sRNA Origin	Target Origin	Target Genes	Function	Reference
miR408	*Puccinia striiformis* f. sp. *tritici* (*Pst*)	*T. aestivum*	*CLP1*	Negatively regulates host immune response by suppressing the expression of *CLP1*.	[92]
*Pst*-milR1	*Pst*	*T. aestivum*	*PR2*	Represses plant innate immune response by suppressing the expression of *PR2*.	[8]
*Pst*-milR1	*Pst*	*T. aestivum*	*SM638*	Innate immunity.	[8]
pt-mil-RNA1	*Pt*	*T. aestivum*	*TCP14*, *CYB5R*, and *EF2*	Suppresses wheat defense response to *Pt* by targeting wheat *TCP14*, *CYB5R* and *EF2*.	[10]
pt-mil-RNA2	*Pt*	*T. aestivum*	*TCP14*, *CYB5R* and *EF2*	Suppresses wheat defense response to *Pt* by targeting wheat *TCP14*, *CYB5R* and *EF2*.	[10]
miR398	*Bgh*	Barley	*HvSOD1*	Negatively regulates host immunity by repressing *HvSOD1* accumulation.	[93]
miR9836	*Bgh*	Barley	*MLA1*	Dampens immune response signaling triggered by host MLA immune receptors.	[94]
*Fg*-sRNA1	*F. graminearum*	Chinese spring wheat	*TaCEBiP*	Suppresses wheat defense response by targeting and silencing *TaCEBiP*.	[90]
*Fol*-milR1	*Fusarium oxysporum*	*Tomato*	*SlyFRG4*	Suppresses host immunity by silencing *SylFRG4*.	[91]
*Osa*-miR167d	*M. oryzae*	Rice	*ARF12*, *WRKY45*	Negatively regulates host immunity by downregulating *AR12* expression.	[95]
miR156	*M. oryzae*	Rice	*SPL14*	Enhances host susceptibility by suppressing the expression of *SPL14* and *WRKY45*.	[96]
*Osa*-miR164a	*M. oryzae*	Rice	*OsNAC60*	Negatively regulates host immunity by suppressing *OsNAC60* expression.	[97]
miR168	*M. oryzae*	Rice	*AGO1*	Negatively regulates host immunity by suppressing *AGO1* expression.	[98]
*Osa*-miR169	*M. oryzae*	Rice	*NF-YAs*	Enhances host susceptibility by suppressing the expression of nuclear factor N-Y (*NF-YA*) genes.	[99]
miR319	*M. oryzae*	Rice	*TCP21*	Negatively regulates host immunity by suppressing *TCP21* expression.	[100]
miR396	*M. oryzae*	Rice	*OsGRFs*	Negatively regulates host immunity by suppressing the expression of *OsGRFs*.	[101]
*Osa*-miR439	*M. oryzae*	Rice	Predicted target genes *LOC_Os01g23940*, *LOC_Os01g36270*, *LOC_Os01g26340* and *LOC_Os06g19250*	Enhances host susceptibility by suppressing the expression of predicted target genes *LOC_Os01g23940*, *LOC_Os01g36270*, *LOC_Os01g26340* and *LOC_Os06g19250*.	[102,103]
miR444b.2	*M. oryzae*	Rice	MADS-box family genes	Negatively regulates host immunity by suppressing the expression of MADS-box family genes.	[104]
siR109944	*Rhizoctonia solani*	Rice	*FBL55*	Suppresses host immunity to sheath blight.	[105]
*Bc*-siR3.2	*Botrytis cinerea* (*B. cinerea*)	*A. thaliana*	*MPK1*, *MPK2*	Suppresses *MPK1*, *MPK2* function in plant immunity.	[88]
*Bc*-siR3.1	*B. cinerea*	*A. thaliana*	*PRXIIF*	Suppresses *PRXIIF* genes.	[88]
*Bc*-siR3.2	*B. cinerea*	*Solanum lycopersicum*	*MAPKKK4*	Suppresses *MAPKKK4* function.	[88]
*Bc*-siR5	*B. cinerea*	*A. thaliana*	*WAK*	Suppression the function *WAK* genes.	[88]
*Bc*-siR37	*B. cinerea*	*A. thaliana*	*WRKY7*, *PMR6* and *FEI2*	Suppresses plant immunity by repressing the expression of *WRKY7*, *PMR6* and *FEI2.*	[89]

*PR2*—Pathogenesis-related 2 gene, *TCP14*—Transcription factor, *CYB5R*—Cytochrome b5 reductase, *EF2*—Elongation factor 2, *TaCEBiP*—Chitin elicitor binding protein, *SlyFRG4*—CBL-interacting protein kinase, *MPK*—Mitogen-activated protein kinases, *MAPKKK*—Mitogen activated protein kinase kinase kinase, *PRXIIF*—peroxiredoxin-2F, *WAK*—Cell wall-associated kinase, *PMR*—powdery mildew resistance, *FEI* (named after the Chinese word corresponding to fat)—leucine-rich repeat receptor-like kinases.

## Data Availability

Not applicable.

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
