# Peer review of "Fungal Secondary Metabolites and Small RNAs Enhance Pathogenicity during Plant-Fungal Pathogen Interactions"

_jof, 2022, doi:10.3390/jof9010004_

Round 1
Reviewer 1 Report
The review article “Fungal secondary metabolites and small RNAs enhance pathogenicity during plant-fungal pathogen interactions” by Mapuranga et al., collected the information on fungal metabolites and small RNAs that are responsible for pathogenicity. This research topic is now attracting attention. Different aspects of different effectors that play role in trigging pathogen infection in plants were reviewed. In addition, different tools for the prediction and study of SM and sRNA effectors were discussed. I think this review article is valuable literature. However, as for tools for the prediction and study of SM and sRNA effectors, more detail explain in figure or scheme is recommended. Other minor point outs are as follows. After considering these revisions, the manuscript can be published.
Words already mentioned in title must not be repeated in keywords
In table 2, the columns entitled as “Cross-kingdom translocation” contained all the same entries as “yes”. So, there is no need of this columns. You can describe this statement in table title or foot note as it is same for all.
Correct the species name, they are written in italic. Also, when a species is cited for the first time in the text, the full text should be written. But afterwards it is abbreviated.
When an acronym is cited for the first time in the text, the full text should be written. After it is only cited using this acronym and not the name complete.
The English language of this manuscript must be revised as there are several grammatical and structural errors throughout the text
Author Response
Response to Reviewer 1 Comments
Point 1: The review article “Fungal secondary metabolites and small RNAs enhance pathogenicity during plant-fungal pathogen interactions” by Mapuranga et al., collected the information on fungal metabolites and small RNAs that are responsible for pathogenicity. This research topic is now attracting attention. Different aspects of different effectors that play role in trigging pathogen infection in plants were reviewed. In addition, different tools for the prediction and study of SM and sRNA effectors were discussed. I think this review article is valuable literature. However, as for tools for the prediction and study of SM and sRNA effectors, more detail explain in figure or scheme is recommended. Other minor point outs are as follows. After considering these revisions, the manuscript can be published.
Response 2: Figure 4 has been added
Point 2: Words already mentioned in title must not be repeated in keywords
Response 2: It has been corrected
Point 3: In table 2, the columns entitled as “Cross-kingdom translocation” contained all the same entries as “yes”. So, there is no need of this columns. You can describe this statement in table title or foot note as it is same for all.
Response 2: It has been deleted since it’s mentioned in the table title
Point 4: Correct the species name, they are written in italic. Also, when a species is cited for the first time in the text, the full text should be written. But afterwards it is abbreviated.
Response 4: All species names have been corrected
Point 5: When an acronym is cited for the first time in the text, the full text should be written. After it is only cited using this acronym and not the name complete.
Response 5: It has been corrected
Point 5: The English language of this manuscript must be revised as there are several grammatical and structural errors throughout the text
Response 5: English language has been revised
Reviewer 2 Report
I consider that the review is suitable for publication in the Journal of Fungi, presenting good definitions of non-protein effectors (secondary metabolites and sRNA), listing examples in relevant pathosystems, as well as some (few) perspectives for potential use in the management of plant diseases .
I only suggest a few spelling corrections:
page 4/line 6: barely>barley
page 4/lines 13, 16, 18: change "Magnaporthe grisea" to the current name: Magnaporthe oryzae.
Author Response
Response to Reviewer 2 Comments
Point 1: page 4/line 6: barely>barley
Response 1: It has been corrected
Point 2: page 4/lines 13, 16, 18: change "Magnaporthe grisea" to the current name: Magnaporthe oryzae.
Response 2: It has been changed